# Recent Progress in Electrochemical Upgrading of Bio-Oil Model Compounds and Bio-Oils to Renewable Fuels and Platform Chemicals

**DOI:** 10.3390/ma16010394

**Published:** 2023-01-01

**Authors:** Jeffrey R. Page, Zachary Manfredi, Stoyan Bliznakov, Julia A. Valla

**Affiliations:** 1Department of Chemical and Biomolecular Engineering, University of Connecticut, 191 Auditorium Rd, Unit 3222, Storrs, CT 06269, USA; 2Center for Clean Energy Engineering, University of Connecticut, 44 Weaver Rd, Unit 5233, Storrs, CT 06269, USA; 3Department of Chemical Engineering, Worcester Polytechnic Institute, 100 Institute Rd, Worcester, MA 01609, USA

**Keywords:** electrocatalysis, biomass conversion, renewable hydrogen, bio-oil upgrading

## Abstract

Sustainable production of renewable carbon-based fuels and chemicals remains a necessary but immense challenge in the fight against climate change. Bio-oil derived from lignocellulosic biomass requires energy-intense upgrading to produce usable fuels or chemicals. Traditional upgrading methods such as hydrodeoxygenation (HDO) require high temperatures (200–400 °C) and 200 bar of external hydrogen. Electrochemical hydrogenation (ECH), on the other hand, operates at low temperatures (<80 °C), ambient pressure, and does not require an external hydrogen source. These environmental and economically favorable conditions make ECH a promising alternative to conventional thermochemical upgrading processes. ECH combines renewable electricity with biomass conversion and harnesses intermediately generated electricity to produce drop-in biofuels. This review aims to summarize recent studies on bio-oil upgrading using ECH focusing on the development of novel catalytic materials and factors impacting ECH efficiency and products. Here, electrode design, reaction temperature, applied overpotential, and electrolytes are analyzed for their impacts on overall ECH performance. We find that through careful reaction optimization and electrode design, ECH reactions can be tailored to be efficient and selective for the production of renewable fuels and chemicals. Preliminary economic and environmental assessments have shown that ECH can be viable alternative to convention upgrading technologies with the potential to reduce CO_2_ emissions by 3 times compared to thermochemical upgrading. While the field of electrochemical upgrading of bio-oil has additional challenges before commercialization, this review finds ECH a promising avenue to produce renewable carbon-based drop-in biofuels. Finally, based on the analyses presented in this review, directions for future research areas and optimization are suggested.

## 1. Introduction

Over the past several decades the utilization of renewable energy has attracted significant interest worldwide [1]. Our consumption of fossil fuels has continued to escalate, depleting non-renewable resources and substantially affecting earth’s climate [2]. Therefore, reducing global fossil fuel usage is one of the most pressing world challenges in the 21st century. Sustainable technologies for generation of electricity from renewable wind and solar energy have seen significant growth in the last several decades. The total solar energy consumption in the United States (U.S.) alone has increased from 120 trillion kJs in 2011 to 1584 trillion kJs in 2021, with projections expecting solar energy to produce 22% of the U.S. total energy generation in 2050 [3]. Nevertheless, there is a significant demand for production of carbon-based commodity chemicals and fuels using renewable energy sources [4,5]. Biomass can be used as an abundant, inexpensive, carbon-based and energy dense feedstock for the production of both renewable fuels and specialty chemicals [5,6]. However, cost effective conversion of biomass into useful products remains a major challenge [7]. Methods for the production of biofuels from lignocellulosic biomass generally fall into one of two categories: bio-chemical (i.e., anaerobic digestion and fermentation) and thermochemical (i.e., hydrothermal liquefaction (HTL), combustion, pyrolysis, and gasification) [5,8,9,10,11]. Thermochemical biofuel production has several advantages over bio-chemical including faster reaction times, increased efficiency, feedstock flexibility and complete biomass utilization [12,13,14].

Fast pyrolysis and HTL target the conversion of biomass to liquid transportation fuels and chemicals. Fast pyrolysis involves the rapid heating (~1000 °C·s^−1^) of biomass to high temperatures (400–600 °C) in the absence of oxygen [15,16,17,18,19,20,21]. HTL converts biomass using liquid water at moderate temperatures (~300 °C) and high pressures (~200 bar) [10,22]. Both technologies target the production of an energy dense bio-oil. The yield and chemical composition of the produced bio-oil depends heavily on the properties of biomass feedstock. Typically, bio-oil contains a large variety of low-carbon number organic compounds such as aldehydes, ketones, carboxylic acids, aromatics and approximately 20 wt% water [23]. Due to the presence of oxygenates, bio-oil has a lower energy content compared to petroleum oil and is prone to polymerization during storage. To make a useable product, bio-oil needs to be stabilized for transportation to a centralized refinery or directly converted into fuels and chemicals. There are several bio-oil upgrading processes including catalytic hydrodeoxygenation (HDO), catalytic cracking, steam reforming, and esterification [12,13,15,19,24,25,26,27,28,29,30]. HDO is the most widely studied upgrading process, wherein the bio-oil is treated with high pressure hydrogen (200–400 bar) and high temperature (200–400 °C) to remove heteroatoms and increase the hydrogen content [31]. Despite its popularity, HDO has several drawbacks including high cost, short catalyst lifetime, consumption of high-pressure hydrogen, and severe reaction conditions [31,32,33].

In addition to the severe process conditions, biomass as a resource is difficult and costly to transport [34,35]. Thus, to make biomass conversion commercially competitive, bio-oil production plans need to be decentralized and small scale. Furthermore, the instability of pyrolysis bio-oil means that in addition to small scale production, bio-oil stabilization or upgrading should occur onsite. To improve biofuel economics and reduce the environmental impact, additional upgrading methods must be explored. High energy inputs required for conventional upgrading processes necessitate development of bio-oil upgrading processes that can be performed effectively at small scales.

Electrochemical upgrading of bio-oils represents a novel avenue for the conversion of bio-oils to usable fuels or commodity chemicals under mild conditions. Several different approaches to electrochemical upgrading of bio-oils have been proposed in the literature, including oxidation and reduction reactions [36,37,38]. Electrochemical hydrogenation/hydrogenolysis (ECH) has been the focus of a growing number of studies as an alternative bio-oil stabilization method [39,40,41,42,43,44,45]. During ECH an oxidation reaction (typically the oxygen evolution reaction) takes place at the positive electrode generating hydrogen ions which are transferred to the negative electrode for the reduction of organic molecules. ECH reactions are intended to stabilized bio-oil by increasing the hydrogen content, and removing oxygen. These reactions can be performed at mild temperatures (<80 °C) and ambient pressures [37]. As the hydrogen used during ECH is generated in situ, no external hydrogen source is needed [37]. In addition, ECH provides a method for coupling the production of biofuels to intermittently generated electricity from solar and wind. The mild nature of ECH could allow for small-scale processing to produce a stable oil that can be easily transported to a centralized (bio)refinery for deep deoxygenation, if needed.

ECH has been tested for the conversion of a wide variety of bio-oil model compounds including benzaldehyde [46], phenol [43], guaiacol [47], and furfural [48]. Although most ECH studies have been performed on model compounds, ECH has been demonstrated on several real bio-oils, finding that it increases both oil stability and hydrogen content. The primary focus of ECH research until today has been the development of efficient and selective electrocatalysts. A wide range of metals have been found to be active for ECH including base (i.e., Ni, Cu, Co) and noble metals (i.e., Pt, Pd, Ru, Rh). Though ECH reactions can be performed at room temperature, high temperatures often result in higher ECH reaction rates [49,50,51,52,53]. Higher applied voltages serve to increase ECH reaction rates, though typically at the expense of efficiency [54]. To date only limited research has been performed on ECH of real bio-oils or even mixtures of bio-oil model compounds. Furthermore, compared to traditional upgrading techniques (i.e., hydrodeoxygenation) the quality of the fuels produced using ECH is much lower. Another major challenge is the low solubility of bio-oils in the aqueous phase [37]. Most studies focused on real bio-oils use either low bio-oil concentrations or the water-soluble fraction of bio-oils [44,45].

This review aims to summarize factors relating to the ECH of bio-oils and its commercialization to identify major research gaps and challenges on the development of ECH of bio-oils. ECH reaction mechanisms are discussed as a basis for the current state of research before a deep dive into factors affecting ECH performance, efficiency, and commercialization. Section 2 of this review focuses on the electrochemical upgrading reactions. We provide the fundamentals of the electrochemical hydrogenation/hydrogenolysis reactions of oxygenates found in bio-oils and discuss the impact of various functional groups on reaction mechanisms. A short description of oxidation and dimerization reactions is also included. Section 3 discusses the factors which effect ECH reactions. Topics covered include electrode design, cell design, temperature, voltage, and electrolytes. Section 4 covers the literature on ECH treatment of real bio-oils. Section 5 focuses on the benefits of ECH from an environmental and economic standpoint, while Section 6 compares thermochemical hydrogenation and electrochemical hydrogenation. Electrochemical upgrading bio-oils and biomass derived compounds has been the subject of several previous literature reviews [36,37,55,56]. Garedew et al. [55] reviewed electrochemical upgrading of depolymerized lignin. Akhade et al. [37] provided a general review on the electrocatalytic hydrogenation of all biomass derived organics. Chen et al. [36] provided reviewed ECH of bio-oils with an focusing on the design and development of ECH systems and ECH reaction mechanisms. This review is intended to provide in depth summary of the factors that influence the electrochemical hydrogenation of bio-oils with an emphasis on the development of novel catalytic materials for the targeted production of renewable fuels and chemicals from ECH.

## 2. Electrochemical Upgrading Reactions

Electrochemical upgrading reactions take three primary forms: electrochemical hydrogenation (ECH), electrochemical oxidation (ECO), and electrocatalytic dimerization (ECD). ECH reactions can be performed at the negative electrode (cathode) using hydrogen ions to increase the hydrogen and reduce the oxygen content in bio-oil molecules [57,58,59,60]. ECO can be performed at the positive electrode (anode) to reduce energy requirement of hydrogen production or coupled with ECH to simultaneously produce value added reduction and oxidation products [43,61]. ECD reactions target the production of bio-oil based polymers [62]. As the primary focus of this review is on ECH of bio-oil model compounds, ECO and ECD reactions will be discussed only briefly.

### 2.1. Electrochemical Hydrogenation/Hydrogenolysis

#### 2.1.1. Fundamentals of the Hydrogen Evolution Reaction (HER) and ECH

Electrochemical reactions in their simplest form require two electrodes (an anode and a cathode) in contact with an ionic conductor. Ionic conductors (or electrolytes) are aqueous solutions of salts, acids, or bases that allow for the transportation of ions between the electrodes of an electrochemical cell. In an electrolytic cell the cathode is connected to the negative pole of a power source and the anode to the positive pole. Hence, the negatively charged ions (the anions) are oxidizing at the anode, while the cations (the positively charge ions) are reducing at the cathode. In an experimental electrochemical cell, the electrode where the reaction of interest is taking place is often referred to as the working electrode (WE), whereas the other is called the counter electrode (CE). The potential at the WE during an electrochemical reaction is measured by using a reference electrode. Reference electrodes have constant known potentials. Some examples often used in the literature include the standard hydrogen electrode (SHE), reversible hydrogen electrode (RHE), Ag/AgCl, and saturated calomel electrode (SCE). The choice of a reference electrode depends on the pH of the electrolyte subjected to investigation.

In water electrolysis hydrogen ions (protons) are generated at the anode, where the water molecules are oxidized at high anodic potentials, and oxygen molecules and protons are formed as products. This reaction is known as oxygen evolution reaction (OER) and is expressed by Equation (1). The protons are transported through an electrolyte and ion exchange membrane to the cathode where the hydrogen evolution reaction (HER, Equation (2)) takes place. Like all cathodic electrochemical reactions, HER proceeds at potentials more negative than its standard redox potential (E^0^). For HER E^0^ is, by definition, 0 V vs. SHE. HER is a multistep reaction, which takes place at the surface of a solid electrode [63] In acidic electrolytes, HER occurs through three elementary reaction steps. The first reaction step is the Volmer step (Equation (3)) in which a hydrated proton associates with an electron and is adsorbed (H_ads_) to the catalyst at an active site “M”. H_ads_ can form H_2_ gas through one of two reaction steps (Equations (4) and (5)). In the Tafel step (Equation (4)) two H_ads_ combine on the catalyst surface to produce H_2_. The Heyrovsky step (Equation (5)) is where a hydrated proton associates with an electron and combines with a H_ads_ to produce H_2_.
(1)$$2H_{2}O\to O_{2}+4H^{+}+4e^{-}E^{0}=1.23\;V,$$
(2)$$2H^{+}+2e^{-}\to H_{2}E^{0}=0\;V,$$
(3)$$H^{+}+e^{-}+M\to {\mathrm{MH}}_{\mathrm{ads}}\;\left(\mathrm{Volmer}\right),$$
(4)$$2{\mathrm{MH}}_{\mathrm{ads}}\to 2M+H_{2}\;\left(\mathrm{Tafel}\right),$$
(5)$$H^{+}+{\mathrm{MH}}_{\mathrm{ads}}+e^{-}\to M+H_{2}\;\left(\mathrm{Heyrovsky}\right),$$
(6)$$C_{6}H_{6}O+6e^{-}+6H^{+}\to C_{6}H_{12}O\;E^{0}=0.15\;V$$

ECH is intended to replace HER at the cathode of an electrochemical cell by integrating the reduction of protons with the hydrogenation of organic molecules. ECH acts essentially as a method for liquid storage of hydrogen produced during electrolysis. ECH reactions are proposed to occur through two general mechanisms; (1) through the reduction of proton to form H_ads_ which can react with an adsorbed organic, or (2) the direct reduction of the organic molecules either through pre-protonation or post-protonation after electron transfer. It is important to mention that under typical ECH operating overpotentials (−0.4 V to −1.1 V vs. Ag/AgCl) the direct reduction of the organic molecules, mechanism (2), cannot take place. To occur, this reaction requires significantly higher overpotentials in comparison to the surface hydrogenation mechanism (1) [64,65,66]. Additionally, at the negative potentials of interest, ECH reactions are competing for catalytic active sites with the HER. However, the E^0^ for most ECH reactions are more positive than that of the HER (>0 V vs. SHE), meaning that ECH reactions are more thermodynamically favorable [37]. The conversion of phenol to cyclohexanol is shown in Equation (6) as an example of a ECH reaction. This reaction involves the transfer of 6 electrons and 6 hydrogen atoms and has an E^0^ of 0.15 V vs. SHE [37]. Despite ECH thermodynamics being more favorable, the kinetics of ECH reactions can be slower than that of the HER. Differences in HER and ECH kinetics can vary significantly based on the electrocatalyst and the composition of the reactants. Faraday efficiency (FE) is often used to measure selectivity of a reaction toward ECH relative to HER. FE is the percentage of electrons used for an electrochemical reaction (in this case ECH) relative to the total electrons passed through the reaction (Equation (7)). Other useful metrics for evaluating ECH performance include specific ECH rate (Equation (8)) specific HER rate (Equation (9)) and turnover frequency (TOF, Equation (10)) [37].
(7)$$\mathrm{FE}=\frac{\mathrm{electrons}\;\mathrm{consumed}\;\mathrm{by}\;\mathrm{hydrogenation}\;\mathrm{of}\;\mathrm{organic}\;\mathrm{compounds}}{\mathrm{total}\;\mathrm{electrons}\;\mathrm{passed}}\times 100\%$$
(8)$$\mathrm{Specific}\;\mathrm{ECH}\;\mathrm{rate}=\frac{\mathrm{moles}\;\mathrm{of}\;\mathrm{reactant}\;\mathrm{consumed}\;}{\mathrm{time}\;\times \;\mathrm{mass}\;\mathrm{of}\;\mathrm{catalyst}\;\times \;\mathrm{metal}\;\mathrm{loading}}\times 100\%$$
(9)$$\mathrm{Specific}\;\mathrm{HER}\;\mathrm{rate}=\frac{\mathrm{moles}\;\mathrm{of}\;\mathrm{hydrogen}\;\mathrm{gas}\;\mathrm{produced}\;}{\mathrm{time}\;\times \;\mathrm{mass}\;\mathrm{of}\;\mathrm{catalyst}\;\times \;\mathrm{metal}\;\mathrm{loading}}\times 100\%$$
(10)$$\mathrm{TOF}=\frac{\mathrm{moles}\;\mathrm{of}\;\mathrm{reactant}\;\mathrm{consumed}\;}{\mathrm{time}\;\times \;\mathrm{dispersion}\;\mathrm{of}\;\mathrm{metal}\;\times \;\mathrm{moles}\;\mathrm{of}\;\mathrm{metal}\;\mathrm{in}\;\mathrm{catalyst}}\times 100\%$$

#### 2.1.2. Impact of Functional Groups on ECH Performance

Bio-oil is an incredibly complex and difficult to study mixture. Most researchers studying electrocatalytic upgrading have utilized model compounds representative of molecules found in bio-oil, which are generally low carbon number aromatic compounds with oxygen containing functional groups (i.e., hydroxyl, methoxy, aldehydes, ketones and carboxylic). ECH has been explored for a wide variety of different bio-oil model compounds [60]. Efficiency and ECH performance depend heavily on the functionality of the organic tested. Figure 1 shows ECH reaction schemes of several bio-oil model compounds studied in the literature.

The effect of hydroxylic groups in organics has been studied extensively through the ECH of phenol. Phenol is one of the most abundant and simple compounds in bio-oils. In fact, the electrochemical reduction of phenol reports back as far as 1914 [67]. Recently there has been a significant interest in increasing the performance and efficiency of phenol ECH. Despite those efforts, there are few reports of direct ECH of phenol into cyclohexane [37,47,68]. Several studies have hypothesized that the activation energy for C-O bond cleavage is significantly higher than that of hydrogenating phenol’s aromatic ring and will only occur at high overpotentials or high temperatures [50,54,69]. Therefore, the majority of researchers have found cyclohexanol as the primary product of phenol ECH, passing through a cyclohexanone intermediate [54,57,58,70]. Song et al. [54] found that during ECH phenol converted to cyclohexanol with 100% selectivity (0.7 V vs. Ag/AgCl, 5% Rh/C electrode, acetic acid buffer pH = 5, room temperature) after 150 min of constant voltage.

The addition of a methoxy group to phenol opens a pathway to hydrogenolysis reactions. Several studies have shown that the ECH of guaiacol results in both the direct hydrogenation product, methoxycyclohexanol, and hydrogenolysis products such as cyclohexanol and cyclohexanone [60,69,70,71]. The major product of guaiacol hydrogenation is typically cyclohexanol as the hydrolysis of the methoxy group is thermodynamically favored in reducing conditions [52,69,70,72,73]. However, as with phenol, cleavage of the C-OH bond (to produce methoxy cyclohexane) is generally not observed. ECH performed on other model compounds containing methoxy groups, such as 3-methoxyanisole and syringol, found that hydrolysis also occurs readily [60,70]. However, the ECH of anisole generally does not result in hydrolysis as cyclohexane is typically not observed [60].

The electrochemical reduction of carbonyls such as benzaldehyde, acetophenone, furfural, hydroxymethylfurfural (HMF), and butyraldehyde have also been intensely studied in the literature [58,65,74,75,76,77,78,79,80,81,82]. Benzaldehyde is readily hydrogenated into benzyl alcohol at high selectivity; however, no further hydrogenation of the aromatic ring has been observed. This is hypothesized to be caused by stearic limitations of the ethyl alcohol group attached to the benzene ring. In select cases, ECH of benzaldehyde can result in the protonation of the phenyl ketyl radical (PhHCO^−^) resulting in the formation of benzaldehyde dimers (hydrobenzoin). This pathway can lead to low efficiency and selectivity towards the desired benzyl alcohol product [65]. Sanyal et al. [78] studied the hydrogenation of benzaldehyde in the presence of varied concentrations of phenol finding that the presence of phenol increased the benzaldehyde reaction rates significantly (reaction conditions: −0.1 V vs. RHE, room temperature, 20 mM benzaldehyde, Pd/C electrode). Sanyal et al. [78] attributed the enhancement of reaction rates to (1) hydrogen bonding interaction with phenol on the surface of the catalyst and (2) proton-coupled electron transfer enabled by adsorbed phenol. Akhade et al. [74] compared the performance of different carbonyl functionalized organics during ECH including benzaldehyde, acetophenone and vanillin finding that the nature of the functional group significantly impacted the ECH rates. Benzaldehyde had the highest ECH rate (1093 µmol g^−1^ s^−1^) and FE (60%), whereas vanillin was completely inactive at the reaction conditions explored (Reaction conditions: Pd/C electrode, 298 K, atmospheric pressure, sodium acetate buffer pH = 5.2, 0.25 V vs. RHE). They proposed that the interaction energy, defined as the lowest energy structure between the organic and electrode as well as organic and solvent, directly correlated to ECH performance. Thus, benzaldehyde had the strongest interaction with the Pd catalyst surface resulting in the fastest ECH rates. The electrocatalytic hydrogenation of furfural has also received significant attention. Furfural is abundant in bio-oil and is readily produced from the acidolysis of hemicelluloses [83,84]. A reaction pathway for the ECH of furfural is shown in Figure 2. ECH of furfural can result in four different products: furfuryl alcohol, 2-methylfuran, hydrofurion, and 1,5-pentidiol [85]. Under most ECH conditions studied in the literature the primary product of furfural hydrogenation is either 2-methylfuran or furfuryl alcohol. Along similar lines ECH of HMF has been the subject of several studies [82,86,87,88]. HMF has two different functional groups (-OH, -C=O) attached to a furan ring. Kwon et al. [88] found that the end products of HMF ECH are 2,5-dihydroxymethylfuran (DHMF), 2,5-dihydroxymethyltetrahydrofuran (DHMTHF), and 2,5-dimethyl-2,3-dihydrofuran (DMDHF).

### 2.2. Electrochemical Oxidation (ECO)

The catalytic oxidation of biomass derived organics is a growing area of research [89,90]. The synthesis of platform chemicals derived from biomass is an attractive concept as it can provide a renewable alternative to petroleum for high value chemicals. Furthermore, performing the oxidation of biomass derivatives via electrochemical route has the potential to produce value added chemicals while simultaneously producing H_2_ [91,92,93,94]. The upgraded organics are more valuable than O_2_ produced during water electrolysis [91]. ECO also avoids much of the safety concerns associated with water electrolysis as hydrogen/oxygen mixtures can be explosive [95]. The kinetics of organic oxidation are faster than that of the OER reaction meaning that the required overpotentials are lower than those for conventional water electrolysis [94]. Figure 3 shows linear sweep voltammograms (LSV) for the oxidation of several different bio-oil model compounds all of which oxidize at lower overpotentials than water [38,95]. The simultaneous production of value-added chemicals and H_2_ has the potential to reduce the energy requirements for H_2_ production by three times [96]. ECO has been demonstrated on a variety of different model compounds including HMF [38,91,92,97], glycerol [98], furfural [85,99], propanol [61], benzaldehyde [100], carboxylic acids [98], and phenol [101]. In particular, the ECO of HMF to furandicarboxylic acid (FDCA) has seen significant attention as FDCA is a building block for polyethylene furanoate (a renewable alternative to polyethylene terephthalate) [38].

Several papers have proposed the simultaneous ECO and ECH of biomass derived organics. This provides a method for producing both oxidation and reduction products simultaneously. A schematic of the combined ECO and ECH of furfural proposed by Zhang et al. [99] is shown in Figure 4. In this case, furfural is oxidized at the anode to produce furonic acid and hydrogenated to furfuryl alcohol at the cathode [99]. The LSV (Figure 4b) shows that the onset potential of the simultaneous ECO and ECH of furfural is decreased from 1.55 V (for water splitting) to 1.2 V. Huang et al. [61] studied the oxidation of 2-propanol combined with the hydrogenation of phenol to cyclohexanol, finding that the applied potential at the anode for 2-propanol oxidation could be limited to 0.2 V rather than the 1.23 V that is required for OER. During simultaneous ECO and ECH the phenol hydrogenation rate was 18 nmol cm^−2^ h^−1^, at 11.4% conversion and 95.5% selectivity to cyclohexanone (3 h, 18.9 mA cm^−2^, 80 °C, anode: 1 M 2-propanol, 4 mg cm^−2^ PtRu/C, 10 mL min^−1^, cathode: 0.1 M phenol, 0.5 mg cm^−2^ Pd/C, 0.1 M phenol). Wu et al. [43] performed simultaneous oxidation of phenol to benzoquinone and hydrogenation of phenol to cyclohexanone. They found that the selectivity to both cyclohexanone and benzoquinone was >99.9% with the respective TOFs of 115.6 and 164.1 h^−1^ and combined FE of 87.7% at 7.5 mA cm^−2^.

### 2.3. Electrochemical Dimerization

Electrochemical dimerization (ECD) is a method for increasing the molar mass of bio-oil while simultaneously removing reactive functional groups (i.e., carbonyls) [62,102,103]. ECD occurs through the generation of organic radicals on the surface of a catalyst [62,102,103]. As both ECH and ECD are performed in reducing environments, their products are usually observed together. Selective dimerization is favorable, particularly using catalysts that have very high overpotentials for HER, since high quantities of adsorbed hydrogen result in increased ECH rates [104]. ECD is relevant for compounds containing carbonyl groups including furfural [62], benzaldehyde [102], acetophenone [105] and HMF [104]. Figure 5 shows a reaction scheme for the dimerization of benzaldehyde and furfural. ECD reactions for carbonyl containing compounds occur through the formation of a ketyl radical [62]. Mixtures of organic molecules with carbonyl functional groups can cross couple as is shown in Figure 5 where benzaldehyde and furfural couple to form 1-(2-furfyl)-2-phenyl-1,2-ethanediol [62].

## 3. Factors That Influence Bio-Oil ECH Reactions

### 3.1. Catalysts for ECH Reactions

The performance, efficiency, and selectivity of ECH reactions depend strongly on the electrocatalyst used. This opens the opportunity to design efficient and selective electrocatalysts to tune the reaction toward specific products. Efficient ECH catalysts will generate adsorbed hydrogen atoms on the surface of the catalyst by the reduction of water (Volmer reaction) but have slow kinetics for H_2_ gas evolution from adsorbed hydrogen thereby allowing for preferential hydrogenation of the bio-oil [99]. As such it is not enough to look at catalysts with high HER overpotentials (i.e., perform poorly in HER) as the HER and ECH reactions are competitive but related. The most studied metals are typically lumped into two categories: (1) platinum group metals and (2) base metals. Typical ECH electrodes are made by one of three methods: spray deposition of carbon supported nanoparticles on to carbon felt [61], metal foils [49], or electrodeposition [106,107].

#### 3.1.1. Platinum Group Metal (PGM) Catalysts

By far the most studied catalysts for electrochemical upgrading reactions are platinum group metal catalysts (PGM). PGM based catalysts show the highest ECH reaction rates and efficiencies. Table 1 shows a summary of ECH experiments performed using PGM catalysts. As it is evident, ECH performance is heavily dependent on the active metals tested and the model compounds studied.

Pt nanoparticles supported on larger carbon nanoparticles (Pt/C) catalyst is by far the most studied catalyst for ECH reactions, since it has been used as a standard in the fuel cell and electrolysis fields [47,72,77,108,109,110,111,112]. Pt is active for the hydrogenation of a wide variety of organic compounds and is active in nearly all conditions tested. For example, Pt catalyst are active for phenol ECH in acid, neutral and basic electrolytes with moderate turnover frequencies (TOF) and current efficiencies [43,57,113,114]. In some cases Pt catalyst has shown to promote C-OH bond cleavage in phenol. However, these reports are relatively rare with most studies reporting the primary products of phenol ECH being cyclohexanol and cyclohexanone [37,51,68,76]. This is similar for other oxygenated organics such as benzaldehyde [65], furfural [112], and guaiacol [71].

Beyond Pt, Pd catalysts have been studied considerably for ECH reactions [50,58,61,65,74,76,80,81,109,111,115,116]. Lercher and coworkers reported several studies on the ECH of benzaldehyde using Pd catalysts [65,76,78,81,117]. Song et al. [65] found that the ECH of benzaldehyde using a Pd/C showed a TOF of 4000 h^−1^ with FE of >99% (acetic acid buffer, pH = 5.2, 0.9 V vs. Ag/AgCl, room temp). Lopez-Ruiz proposed that the high performance of benzaldehyde ECH is related to the optimal binding energy of benzaldehyde on the Pd surface [111]. Using DFT-based ab-initio molecular dynamics (AIMD) calculations, Akhade et al. [74] found that the aromatic ring of the benzaldehyde molecule is attracted to the charged Pd surface, whereas acetophenone and vanillin were repelled. Selectivity toward the ECH reaction can be a problem for benzaldehyde on Pd catalysts, and depending on the reaction conditions significant production of hydrobenzoin (benzaldehyde dimer) may occur [62,79]. The performance of Pd catalysts for other model compounds can vary significantly [58,109]. Phenol ECH with Pd catalysts has been shown to be active only in neutral and basic electrolytes [76,118]. For acidic electrolytes phenol ECH using Pd catalysts results in nearly no reaction [65]. Singh et al. [76] proposed that in acidic electrolytes the Pd surface is saturated with adsorbed hydrogen preventing phenol hydrogenation.

Ru and Rh have been studied as ECH catalysts less than Pt or Pd [47,52,109,110,119,120]. Li et al. [52] studied ECH of phenol, guaiacol, and syringol using a Ru/ACC (activated carbon cloth) electrode, and found that the Ru catalyst was active for all three phenolic compounds. Gardew et al. [60] screened the impact of functional group types, positions, and sizes, on product conversion and selectivity using Ru/ACC electrode and revealed that the catalyst was active for ECH of phenol, p-cresol, 4-ethylphenol, and 4-propylphenol to their corresponding alkyl-cyclohexanols. However, studies have shown that Ru catalysts promote HER over ECH reactions, when benzaldehyde, furfural and heptanal were used as model compounds, leading to low turnover frequencies and FEs compared to other PGM group catalysts [111]. In general, most of the studies agree that Rh catalysts are particularly active for ECH of phenol and outperform Pt and Pd under the same reaction conditions [57,58].

The natural progression of catalyst design is to go beyond single metal catalysts to bimetallic and trimetallic catalysts to improve performance and selectivity. Several different bimetallic and trimetallic catalysts have been tested for ECH reactions [43,48,73,77,87,121,122,123]. For example, Wu et al. [43] screened 22 different single and bimetallic catalysts for their performance in ECH of phenol. The reaction rates followed the trend of PtRu > FePt > RhPt > NiPt (TOF h^−1^: 213.5, 170.9, 143.9, 115.1). However, as their target product was cyclohexanone further studies were conducted on the NiPt as it displayed the highest selectivity to cyclohexanone. NiPt hydrogenated phenol to cyclohexanone at >99% selectivity and nearly 90% efficiency (reaction conditions: H_2_SO_4_, 25 °C, −0.05 V vs. RHE, 3 h reaction time). The PtNi alloy showed excellent performance and selectivity to cyclohexanone, suppressing hydrogenation of the carbonyl group. Wu et al. [43] proposed that the electron transfer from Ni to Pt allows the Pt to become more electronegative and prevent absorption of cyclohexanone onto the metal’s surface. Similarly, Zhou et al. [108] studied the performance of PtNi and PtNiB alloys on the hydrogenation of guaiacol, phenol and several other organics. Both the PtNi/CMK-3 and PtNiB/CMK-3 catalysts showed excellent selectivity to cyclohexanol using both phenol and guaiacol. Guaiacol was hydrogenated to cyclohexanol with a 90.3% and 90.8% FE (Reaction conditions: 60 C, 40 mA constant current, 0.2 M HClO_4,_ 90 min reaction time). Figure 6d shows the effect of metal and support on guaiacol ECH, where PtNiB/CMK-3 catalyst clearly demonstrated the fastest reaction rates compared to Pt/CMK-3 and commercial Pt/C. Figure 6a,b shows polarization curves for the PtNi/CMK-3 and PtNiB/CMK-3 catalysts. For PtNiB/CMK-3 (Figure 6a) the polarization curve in the presence of guaiacol shows a significantly lower onset potential (−0.26 V vs. RHE) than the polarization curve without guaiacol (−0.41 V vs. RHE). This implies that the ECH reaction of guaiacol is more favorable than the HER, using PtNiB. These results were further investigated using DFT calculations. The presence of B atoms in the PtNi alloy significantly increased the binding energy of guaiacol onto the surface (−0.59 eV vs. −0.91 eV) which contributes to the improved performance.

As previously discussed, the ECH of guaiacol and other methoxyphenols on PGM catalysts leads to significant production of cyclohexanol and cyclohexanone as the hydrolysis of the -OCH_3_ bond is significantly favored over the cleavage of the -OH bond [52,60,72,108]. Limiting the hydrolysis of the -OCH_3_ group is one potential avenue to value added chemicals as the methoxylated cyclohexanols are valuable feedstock chemicals in the pharmaceutical industry [72,124]. Recently trimetallic electrodes PtRhAu [72] and RhPtRu [73] have shown to significantly boost the selectivity and efficiency for the hydrogenation of guaiacol to 2-methoxycyclohexanol (2MC). Peng et al. [72] used DFT calculations to show that the addition of Rh and Au to the platinum catalyst increases the surface coverage of guaiacol and decreases the -OCH_3_ bond length suppressing hydrolysis and favoring hydrogenation. Additionally, experimental results from Peng et al. [72] supported the theoretical results reporting a 60% FE toward the production of 2MC (Reaction conditions: 0.2 M HClO_4_, 20 °C, 200 mA cm^−2^, 1 h reaction time). Figure 7 shows results from Wang et al. [73] for the ECH of guaiacol using RhPtRu catalyst. The RhPtRu catalyst also significantly increased the production of 2MC with 50% Faradaic efficiency and 60% selectivity to 2MC after one hour of reaction. The RhPtRu catalyst also showed significant increase in guaiacol ECH performance compared to other single metal catalysts found in the literature (Figure 7d).

Several bimetallic Pd alloy catalysts have shown promise for ECH reactions, particularly for the ECH of benzaldehyde and furfural [48,77,121,125,126]. Brosnahan et al. [48] found that the performance of AgPd alloy showed excellent selectivity toward the production of furfuryl alcohol reducing both coupling and hydrogenolysis reactions. The FE was >95% at 0.5 V vs. RHE and the selectivity to furfuryl alcohol was nearly 100% (Reaction conditions:100 mM furfural, room temperature, potassium buffer). Zhou et al. [126] tested the performance of PdCu catalyst for the production of methylfuran from furfural. PdCu/C catalysts were synthesized using the metal organic framework HKUST-1 as a sacrificial template to create a porous electrically conductive material. HKUST-1 was pyrolyzed at 800 °C to produce a Cu/C porous framework and then the catalyst was subjected to galvanic replacement of Cu with Pd to create a CuPd alloy. A schematic for the production of PdCu/C catalysts from Zhou et al. [126] is shown in Figure 8. The electrodes were then tested for the ECH of furfural showing that the addition of Pd resulted in a significant increase in methylfuran production. The Faradaic Efficiency of methylfuran using CuPd alloy was 75% at 0.58 V vs. RHE (Reaction conditions: 0.1 M acetic acid buffer, 4.5 mA cm^−2^, 20 °C). Wu et al. [46] tested a dendrite like PdCu/Cu foam catalyst for several biomass derived aldehyde compounds including benzaldehyde, vanillin, carboxyl benzaldehyde and syringaldehyde, finding that the aldehydes were hydrogenated to their alcohol constituent at FE greater than 90% and alcohol selectivity greater than 95% (Reaction conditions: 0.3 V vs. RHE, 0.1 M H_2_SO_4_, room temperature).

Li et al. studied single atom alloy Ru_1_Cu catalyst for the hydrogenation of HMF to 2,5-dihydroxymethylfuran (DHMF) [87]. A schematic of the process for producing atomically dispersed Ru_1_Cu is shown in Figure 9. Ru was atomically dispersed on a Cu foam support. Experiments were performed for atomically dispersed Ru, Ru nanoparticles, and the Cu foam by itself. When compared to the Cu support by itself the Ru_1_Cu catalyst produces higher ECH efficiencies (85.6% vs. 71.3%, at −0.3 V vs. RHE) at faster production rates (0.47 mmol cm^−2^ h^−1^ vs. 0.08 mmol cm^−2^ h^−1^, at 0.3 V vs. RHE). The Ru served the reduction of the overpotential required for the ECH of HMF, retaining the selectivity over the HER reaction, which would typically be seen using Cu as catalyst. The single atom catalyst performed better than the nanoparticle catalyst owing to the lower HER overpotential of the latter and therefore higher selectivity to H_2_ production. The single atom catalyst also limited the production of dimer product (bis(hydroxymethyl)hydrofuroin), which was a significant problem for the Cu catalysts. This selectivity to DHMF was maintained up to HMF concentrations of 100 mM.

#### 3.1.2. Non-PGM Catalysts

Non-PGM catalysts have garnered significant research interest as inexpensive alternatives to PGM catalysts. Remarkably, a large number of non-PGM catalysts have shown good activity for the ECH of oxygenated organics [42,49,111]. This is likely because the applied current during ECH keeps the base metal in the reduced active state [111]. This differs from thermochemical hydrogenation/hydrodeoxygenation process, since the source of hydrogen for ECH reactions is the reduction of an electrolyte, rather than the dissociation of H_2_. Non-PGM catalysts that showed activity for the ECH of bio-oil model compounds are based on transition metals, such as: Ni, Cu, Zn, Co, Fe, Pb, and Hg [70,112,127,128,129].

In general, reaction rates using non-PGM catalysts are lower than those using PGM catalysts, especially at low potentials. By far the most studied non-PGM catalysts for ECH reactions are based on Ni and Cu [41,44,65,85,116,119,130,131,132]. Ni catalyst has moderate performance for the ECH of several model compounds such as phenol [130], guaiacol [130], furfural [116] and benzaldehyde [65,116,130]. In some cases, the performance of Ni catalyst was better than that of PGM catalysts. For example, Song et al. [65] compared the ECH of benzaldehyde using Ni/C, Pd/C, Pt/C and Rh/C and found that the Ni/C catalyst resulted at the highest TOF (4730 h^−1^, 3899 h^−1^, 2267 h^−1^ and 2189 h^−1^, respectively, reaction conditions: acetic acid buffer pH = 5, 0.9 V vs. Ag/AgCl, room temperature). However, when ECH of benzaldehyde was performed at lower voltages the PGM catalysts showed higher TOFs. The conversion of aryl ethers is highly selective for C-O bond cleavage over ring hydrogenation using Ni catalysts in alkaline conditions [41,133]. Cu catalysts have seen significant attention for ECH of furfural, [40,49,85,134,135,136,137,138] having demonstrated very high selectivity to 2-methylfuran (~60%) in highly acidic electrolytes [49,134]. Their major draw backs include leaching of Cu in acidic conditions and furfural dimerization [49,116].

Studies comparing the performance of different base metal catalysts under the same reaction conditions are limited, thus it is difficult to make meaningful comparisons. Lam et al. [70] studied the performance of several different Raney type catalysts including Raney-Ni, Devarda-Cu, and Raney-Co for the ECH of guaiacol. They found that Raney-Ni performed better, while Raney-Co showed only traces of hydrogenation products and Devarda-Cu did not produce noticeable products. Andrews et al. [116] compared Cu, Ni, Zn, and Co catalysts for the hydrogenation of benzaldehyde finding that Cu catalyst has the highest activity for both ECH and HER. The reported ECH activity order for these catalysts is Cu > Ni > Zn > Co. Further experiments with furfural, heptanal, cyclohexane-carboxaldehyde, and acetophenone found that the performance of non-PGM catalyst depended heavily on the model compound tested. Koper and coworkers compared the performance of different metals on the ECH of HMF in both neutral [139] and acidic [88] conditions. For acidic conditions Kwon et al. [88] tested a wide variety of catalysts, including Fe, Ni, Cu, Pb, Co, Ag, Au, Cd, Sb, Bi, Pd, Pt, Al, Zn, In, and Sb. The authors reported that Ni catalyst possess the highest selectivity to DHMF (hydration of the carbonyl group to an alcohol), and the Sb catalyst having the highest selectivity to DMDHF (hydrogenolysis of the carbonyl and hydroxyl group of HMF).

Researchers have also studied bimetallic, non-PGM catalysts. Several researchers have reported improved ECH performance using AgCu catalysts [107,122]. Both Sanghez de Luna et al. [107] and Li et al. [122] showed significant improvement on the selectivity of 5-hydroxymethylfurfural (HMF) ECH to 2,5-bis(hydroxymethyl)furan (BHMF) using AgCu catalysts. Sanghez de Luna et al. [107] tested AgCu catalyst that has been synthesized through both galvanic replacement and electrodeposition of Ag onto a Cu foam. ECH experiments with HMF as a model compound showed that the bimetallic AgCu catalyst performed significantly better than a bare Ag or Cu foam. Similarly, Li et al. [122] synthesized AgCu alloyed nano particles supported on a pyrolyzed biomass alginic acid sodium. The AgCu catalyst performed very well for ECH of HMF with 94% selectivity toward BHMF at 93.2% conversion and 61.8% FE, (Reaction conditions: −0.34 V vs. RHE, 0.05 M Na_2_B_4_O_7_, room temperature).

Some recent studies have shown that non-precious metal phosphide catalysts can be excellent ECH catalysts [99,140] Zhang et al. [99] studied the performance of Ni_2_P and Cu_2_P catalysts for the ECH of furfural to furfuryl alcohol. Figure 10 shows results from the ECH of furfural using carbon fiber cloth (CFC), Cu/CFC, Ni_2_P/CFC, Pt/CFC and Cu_2_P/CFC. Among these catalysts, Cu_2_P/CFC catalyst exhibits the highest conversion (92–98%) and selectivity (~100%) under potential range of −0.05 V to −0.55 V (Reaction conditions: KOH, 50 mM Furfural). Yang et al. [140] tested the performance of a FeP-MoP-Fe foam electrode for the ECH of nitrobenzyl alcohol to 4-aminobenzyl alcohol, finding that the electrode greatly improved the conductivity and electron transfer resulting in faster kinetics and superior 4-aminobenzyl alcohol selectivity compared to FeP and MoP by themselves.

#### 3.1.3. Electrode and Catalyst Support Materials

The electrode support plays a crucial role on ECH performance. Generally high surface area conductive materials are desired for high performance electrodes. Carbon materials such as nitrogen-doped hierarchically porous carbon [43], activated carbon cloth [52,60,120], graphite [68], carbon felt [73,80,141], carbon fiber cloth [99], carbon black [43,113,123] and ordered mesoporous carbon [108,119] have positive effect on the catalytic performance. In cases where an electron transfer catalyst (ie. Polyoxometalate) was used, insulating materials, such as aluminum oxide (Al_2_O_3_), have also shown to be active [47].

Metal nanoparticles supported on carbon black remain some of the most popular electrode supports. The highly dispersed nature of metal on carbon black gives it a high electrochemically active surface area and reduces the required metal loading compared to foil-based electrodes. These electrodes are often a variation of those used in PEM fuel cells or electrolyzers in which metal nanoparticles supported on carbon black bound to ion exchange membrane using Nafion^®^. Amouzegar and Savadogo [97] studied the performance of highly dispersed Pt nanoparticles supported with Vulcan carbon XC-72 (Cabot Sarnia, Canada) compared to platinized platinum (Pt/Pt). The dispersed particle showed significant improvement over the Pt/Pt catalyst with the authors finding that the FE for phenol hydrogenation improved from 8% using the Pt/Pt catalyst to 70% for 30 wt% Pt-C. Further decreases in Pt loading showed no significant improvement in FE, despite having significantly higher electrochemically active surface areas. In recent years, carbon black supports have served as a baseline for the development of novel carbon materials for improved electrode performance [43]. For example, Wu et al. [43] compared the performance of Pt supported on carbon black with nitrogen-doped hierarchically porous carbon finding that the TOF increased from 77.7 h^−1^ to 83.4 h^−1^.

Several studies have investigated the effect of the pretreatment of carbon supports on the performance of ECH reactions. Carbons can be treated using a variety of methods to tune the surface containing functional groups including thermal, acid treatment, O_2_, ozone, and plasma. Xu et al. [42] studied the effect of calcination temperatures of carbon on the performance of the nitrogen doped carbon supports for furfural hydrogenation. Three different calcination temperatures were tested including 900, 800, and 700 °C. They found that the higher calcination temperature has positive effect on the hydrogenation reactions with the highest calcination temperature (900 °C) producing the highest yield (99%) conversion (98%) and selectivity (99%) to furfural. The authors attributed this increase in performance to the higher degree of graphitization and therefore higher conductivity. Koh et al. [117] studied the performance of various carbon supports, which were subjected to different pretreatment methods including thermal oxidation, nitric acid oxidation and plasma oxidation. They found that the performance of benzaldehyde ECH was related to the acid content in the support. The plasma treated carbon fiber showed the best performance with TOF of 3000 h^−1^ compared to ~500 h^−1^ of the untreated carbon fiber. Chu et al. [121] synthesized a triple junction ZrO_2_/Pd/C catalyst to mimic the functionality of redox enzymes. The addition of ZrO_2_ increased the TOF in benzaldehyde ECH by 200% (Figure 11). They proposed that the presence of ZrO_2_ increases the local acidity around the Pd active sites and enhances the proton/electron transfer properties of the Pd.

### 3.2. Electrochemical Cell Designs

This section provides an overview of electrochemical cell designs used in the literature for ECH. These cells are typically divided into two categories: (1) experimental electrochemical cells and (2) flow type cells. The main representative of the first category is so called electrochemical “H-cell”. The flow type cells are variation of single cell hardware that is used in testing fuel cells. In this type of cells, the reactant solutions are delivered to the porous catalysts layers by flowing through specially designed channels in the flow plates of the cell hardware. This section is intended tο provide an overview of the relevant designs used in the literature.

#### 3.2.1. Experimental Electrochemical Cell

Experimental electrochemical cells for ECH are typically separated into two categories: (1) H-cell, and (2) undivided cell [59,63,101]. The undivided cell consists of a single compartment with three electrodes (working, auxiliary and reference). Without a divider, reactants can mix throughout the cell for both hydrogenation at the cathode and oxidation at the anode [45,142]. To avoid unwanted oxidation or reduction most studies employ an H-cell, which consists of two compartments separated by either a glass frit or an ion exchange membrane [49,112,140]. H-cells are simple batch systems, which allow for fundamental exploration of reactions, rapid catalyst testing, accurate measurement of potentials, as well as for study of reaction kinetics without temperature or concentration gradients. One advantage of two compartment cells is that the anode and cathode can operate at different conditions. (i.e., different electrolytes or coupled ECH and ECO) to optimize reaction performance. Most studies use a stir bar to maintain a well-mixed solution [71]. An inert gas such as nitrogen or argon can be used to remove dissolved oxygen from the aqueous solution and allow for the collection of water insoluble products [49,134,143]. In cases where an inert gas is used to purge the H-cell, it may be desirable to attach a condenser and cold trap like the example shown in Figure 12. Another important design consideration is the distance between the anode and cathode, which should be minimized to reduce ohmic resistances associated with the long distances between electrodes [54,65].

A modification of the H-cell allows to suspend a catalyst in solution and create a so called “stirred slurry electrocatalytic reactor” (SSER) [47,69,71,115]. As mentioned in the previous section most electrocatalytic upgrading studies have been performed with catalysts fixed onto a solid surface (i.e., carbon felt), in which the electrode is prepared using spay deposition, electrodeposition or impregnation methods. Recent research has shown that slurry reactor systems may be a promising alternative [40,47,69,71,115,144]. In this case, catalyst particles are suspended in the electrolyte solution and the reaction occurs when catalyst particles collide with the current collector (typically a graphite rod). The primary advantage of this type of system is the improved mass transfer between the reactants and the catalyst. Wijaya et al. [71] tested the performance of guaiacol ECH in an SSER reactor finding that complete guaiacol conversion was achieved after 4 h at 165 mA cm^−2^ with a FE of 78% (Reaction conditions: 5 wt% Pt/C, methylsulfonic acid electrolyte, 40 °C). Wijaya et al. [71] also found that the stirring rate is a key parameter for SSER reactors as it affects the adsorption of the reactants with the catalyst and collisions of the catalyst with the current collector.

As mentioned above, most slurry systems rely on the collision of catalyst particles and a current collector. This process is electrically inefficient and cannot significantly improve the electrocatalytic performance. Several recent studies have investigated the addition of a soluble polyoxometalate (POM) as an electron transfer catalyst [47,144]. In this dual-catalyst system, the POM functions as a charge storage and a transfer catalyst which greatly improves the charge transfer efficiency of the stirred slurry system. Figure 13 shows a schematic of the dual-catalyst slurry system proposed by Liu et al. [47]. Liu et al. [47] tested the performance of SiW_12_ with Pt/C suspended catalyst for ECH of several bio-oil model compounds including phenol and guaiacol. For guaiacol hydrogenation the dual-catalyst system (SiW_12_-Pt/C) achieved >99% FE at current densities up to 800 mA/cm^2^. In addition, the SiW_12_ improved the selectivity of phenol and guaiacol to cyclohexane reaching 18% selectivity. The POM is proposed to catalyze the protonation of the C-O bond allowing for improved cyclohexane production. DFT studies showed that the activation energy of the C-O cleavage is significantly reduced when phenol is solubilized by the SiW_12_ instead of water. Zhai et al. [144] employed a similar dual-catalyst system for the hydrogenation of 2-methoxy-4-propylphenol (MPP) as a model compound for lignin oils. In their system NaBH_4_ was added as a reducing agent for the POM. NaBH_4_ reduced and drove PW_12_ to a negative redox potential in the solution allowing for hydrogenation of MPP to occur either thermally or electrocatalytically. Several different POMs have been tested for the ECH of MPP, including SiW_12_, PW_12_, PMo_12_, and SiMo_12_. PW_12_ showed the best performance allowing for 60% selectivity to the deoxygenated product propylcyclohexane at 50 mA cm^−2^ and FE 90% (reaction conditions 5 wt% Pt/C, 80 °C, 30 min reaction time) [144].

#### 3.2.2. Flow Type Cells

Flow cells are preferred for chemical synthesis of large-scale products, such as fuels because they can significantly increase production rates compared to batch systems. The most commonly used flow cells for electrochemical upgrading of bio-oils and bio-oil model compounds are adopted from the fuel cells research. An example of an electrochemical flow type cell is shown in Figure 14. These cells consist of the following several main components: flow channels, a current collector, gas diffusion layer and a catalyst coated membrane (CCM). Variety of different catalysts and feeds [61,145,146,147,148,149,150,151,152] have been used in flow type cells to study ECH [45,127,128,129,130,131,132,133,134] of phenol [61,108], guaiacol [108], benzaldehyde [146], furfural [151], acetophenone [149], levulinic acid [147] and acetone [150]. The direct coating of the membrane with the catalyst results in significantly reduced electrical resistance. However, the catalyst in these CCMs is often deposited as a thin surface coating bound to the membrane with a conductive polymer, which can make diffusion of the liquid phase reactants challenging [145]. Turbulence promoters such as a gas diffusion layer are often used to increase the mass transfer coefficients of the reactants. Feeding pure or diluted hydrogen into the anode side of the reactor is a modification of the flow type cell, which has garnered some attention [145]. As a result of this modification, the potential of the half-cell reaction at the anode of such a cell is zero, and the negative effect of any anodic oxidation reaction on the ECH at the cathode is eliminated.

Other approaches for electrochemical flow cells include flow-through cells and dual membrane cells. Figure 15 shows an example of a dual membrane cell used for electrochemical hydrogenation of pyrolysis oil proposed by Lister et al. [153] Τhis cell has two membranes, the cation exchange membrane (CEM) and the anion exchange membrane (AEM). Protons are generated at the anode, transported through an electrolyte solution, which separates the two membranes, and is used at the cathode for the ECH reaction. The advantage of this type of cell is that small organic acids, which lower the pH of bio-oils, can pass through the AEM reducing the acid content of the bio-oil [153]. Fixed bed systems like the one proposed by Lopez-Ruiz et al. [80] are designed to improve mixing of reactants in the flow cells. The cell designed by Lopez-Ruiz allows the electrolyte and organics to flow through the catalyst bed, an impregnated carbon felt, improving mixing of reactants and maintaining good electrical conductivity.

### 3.3. Effect of Temperature

One advantage of electrochemical upgrading is that it operates at lower temperatures (20–80 °C) compared to traditional thermochemical upgrading processes reducing the required energy input [58,111,134,136,153]. However, increasing temperature generally serves to increase the reaction rate of the ECH reactions [49,50,51,52,53]. Song et al. [50] found that the conversion of phenol using Pt/C increased with temperature up to 50 °C. At temperatures higher than 50 °C the catalyst deactivated rapidly. Singh et al. [51] later proposed that the deactivation is a result of dehydrogenated phenol species, strongly bound on the Pt/C surface. In addition to reaction rates temperature can also affect the reaction selectivity and FE. Li et al. [52] studied the effect of temperature on ECH of guaiacol finding that at 25 °C and 50 °C 2-methoxycyclohexanol was the dominant product but at 80 °C the selectivity of cyclohexanol was greater than 60%. The FE was 17% at 50 °C, higher than at 25 °C or 80 °C (8% and 10%, respectively).

### 3.4. Effect of Voltage and Current

Potential, and by extension current, plays a crucial role in ECH as they are the parameters that allow electrochemical reactions to occur. The standard state equilibrium potential of an electrochemical reaction (E°) represents the potential in which redox pairs are at equilibrium. At E° no reaction occurs. Electrochemical reactions are driven to completion when a current is passing through the cell. In the case of ECH, the water oxidation reaction is taking place at the anode, while the hydrogenation of organics is occurring at the cathode. ECH reactions typically run at either constant current (galvanostatic mode) or constant potential (potentiostatic mode). Both applied potential (V vs. RHE) and surface area normalized current (i.e., mA cm^−2^) data are needed for meaningful comparisons among the studies reported in the literature. Higher applied voltage (or current) generally serves to increase ECH rate by increasing the available hydrogen to hydrogenate the organics of interest. However, this increase is usually accompanied by a loss in FE. Typically, the FE increases to an optimal current density (or voltage) before it begins decreasing. Song et al. [54] found that for phenol ECH on a Rh/C electrode the FE increased from 20% at −0.4 V vs. Ag/AgCl (0.02 mA cm^−2^) to 70% at −0.7 V vs. Ag/AgCl (0.16 mA cm^−2^), but a further increase to −0.9 V vs. Ag/AgCl (0.25 mA cm^−2^) resulted in an efficiency decrease to 66%. However, ECH rate was the highest at −0.9 V with TOF of 629 h^−1^ compared to a 475 h^−1^ at −0.7 V.

### 3.5. Effect of Electrolyte

The electrolyte plays a key role, because the reduction of electrolyte on the catalyst surface provides the protons for ECH reactions. Hence, the choice of electrolyte can affect the reaction rate, efficiency, and product selectivity. In some severe cases the choice of electrolyte can completely quench ECH. For example, phenol ECH using Pd electrodes in acidic electrolytes results in near zero reaction rates owning to complete surface coverage of adsorbed hydrogen [65]. The typical electrolytes used in ECH reactions are grouped based on their pH. Some examples of electrolytes used in the literature include basic electrolytes such as NaOH and KOH, neutral electrolytes such as NaCl, and acidic electrolytes such as H_2_SO_4_, HCl, and HClO_4_ [52].

As a general trend, most ECH studies use acidic or neutral electrolytes because they result in significantly higher reaction rates compared to basic electrolytes. Acid and neutral electrolytes are proposed to be the best for guaiacol ECH. The ionic state of guaiacol changes with the pH of the solution (pKa of 9.9). As a result, in neutral and acidic conditions guaiacol remains a neutral molecule but in basic conditions a phenolate ion forms. The ion prefers to remain in the polar aqueous phase, reducing significantly the adsorption onto the catalyst surface. Wijaya et al. [69] studied the effect of catholyte-anolyte pairs on guaiacol and phenol ECH in a SSER reactor. They evaluated acid-acid (H_2_SO_4_–H_2_SO_4_), neutral-acid (NaCl–H_2_SO_4_), base-acid (H_2_SO_4_–NaOH), and acid-base (H_2_SO_4_–NaOH) finding that all combinations of catholyte-anolyte pairs were active for guaiacol and phenol ECH except base-acid. The neutral-acid pair produced the highest FE (94%) but slightly lower conversion than the acid-acid pair (36.36% vs. 37.88%). The selectivity of the reaction also changed based on the electrolyte. The major product for the acid-acid pair was cyclohexanol, whereas 2-methoxycyclohexanol was the major product when the neutral-acid pair was used. Similarly, the pH of the electrolyte plays a significant role on furfural ECH and can drive the reaction to three different products: pH below 3 favors methyl-furan production, pH between 3 and 9 favors furfuryl alcohol, and pH higher than 9 favors dimerization [137,143,154]. Jung and Biddinger [137] found that using a copper flag electrode at a pH of 3.4 (0.2 M NH_4_Cl) during ECH of furfural, resulted in methylfuran selectivity lower that 2%. Further decrease of the pH to 1.1 (0.1 M H_2_SO_4_) resulted in methylfuran selectivity of 20%, and at pH 0 the methylfuran selectivity was 31% (Reaction conditions: 10 mA cm^−2^, 20 vol% acetonitrile in DI water, room temperature). In the case of benzaldehyde several studies have shown that alkaline electrolytes favor coupling reactions rather than ECH [62,79]. Birkett and Kuhn [79] studied ECH of benzaldehyde in 1 M H_2_SO_4_, 1 M Na_2_SO_4_ (pH = 7), and 2 M NaOH finding that the alkaline electrolyte (NaOH) produced the highest ratio of hydrobenzoin/benzyl alcohol (5.67) compared to neutral (Na_2_SO_4_, 1.97) and acidic (H_2_SO_4_, 0.83) (Reaction conditions, Pb cathode, 10 mA cm^−2^, 30 min reaction time).

## 4. Electrochemical Upgrading of Real Oils

Published research on electrocatalytic upgrading of real bio-oils remains scarce. The complex nature of real bio-oils makes them difficult to study. There are several challenges working with real oils such as closing the mass balance, quantitative and qualitative identification of products and solubility of the oils. Real bio-oils studied in the literature are derived from either fast pyrolysis of lignocellulosic biomass or lignin depolymerization [44,45,59,119,144,153,155]. Li et al. [155] studied ECH of water-soluble bio-oil (WSOB) derived from the fast pyrolysis of poplar using a Ru/ACC electrode. They found that ECH can convert most of the ketones and aldehydes into their corresponding alcohols. Accelerated aging tests showed that ECH can increase the stability of the bio-oil by preventing polymerization. Ethylene glycol and propylene glycol were the major products with a carbon yield of 10%. The addition of a cationic surfactant cetyltrimethylammonium bromide (CTAB) improved the electrical efficiency from 4% to 10% after 6.5 h of ECH. Zhang et al.[119] studied the ECH of WSOB derived from fast pyrolysis of corn stover. They found that ECH promoted the production of aliphatic protons and significantly increased the content of hydroxyl functional groups in the bio-oil (particularly polyhydric alcohols). Andrews et al. [116] tested the performance of Cu and Pd catalysts on the electrocatalytic hydrogenation of pyrolysis oil. 80 mM of benzaldehyde was co-fed with 0, 1, 10 wt%. of bio-oil derived from pyrolysis of pinewood. Both catalysts were initially active for ECH. However, Cu catalyst showed a significant increase in half cell potential after the reaction (~ 1.0 V vs. Ag/AgCl before reaction 2.0 V vs. Ag/AgCl after ECH) implying that the Cu catalyst deactivated during ECH. Lister et al. [153] tested the upgrading of pyrolysis oil using a dual membrane flow cell (described in detail in Section 3.2.2). The dual membrane flow cell system allowed the separation and removal of small carboxylic acid compounds (i.e., acetic acid and formic acid) from the bio-oil. Treatment of fast pyrolysis oil from lower resin wood and ponderosa pine showed reduction in the carboxylic acid number (from 76.4 to 33.9 after ECH treatment with a total charge of 1306 coulombs (C)) and total acid number (from 192.1 to 166.6 after ECH with a total charge of 1306 C).

Wang and coworkers have performed several studies on the electrocatalytic upgrading of fast pyrolysis oil in an undivided H-cell and a platinum foil cathode [44,45,59,142]. Deng et al. [59] studied ECH of a water-soluble bio-oil fraction, whole bio-oil and dichloromethane extracted fraction of rice husk pyrolysis oil. They found that the phenol content of the bio-oil reduced significantly over time. To assist with the complex nature of pyrolysis oil Fourier transform ion cyclotron resonance mass spectrometry (FT-ICR MS) was used to show that the bio-oil underwent both hydrogenation reactions and condensation reactions [59]. Wang et al. [142] and Deng et al. [45] studied the formation of coke on the anode during electrochemical upgrading of rice husk pyrolysis oil in an undivided cell. They found that radicals form on the surface of the anode by the oxidation reaction, lowering the efficiency of the process and reducing the desired hydrogenation reactions. Similarly, Deng et al. [44] studied polymerization during ECH demonstrating that large molecules (>800 Da) polymerize, and the polymerization is initiated at the cathode. Increasing current density can increase the rate of polymerization and the rate of polymerization can surpass the rate of coke formation.

Zhai et al. [144] studied the electrochemical upgrading of lignin oil derived from the catalytic depolymerization of poplar, pine and cornstalk lignin. Catalytic depolymerization was performed in methanol using Ru/C catalyst. The resulting oil which contained primarily oxygenated lignin monomers (mostly 2-methoxy-4-propylphenol and 2,6-dimethoxy-4-propylphenol) was subjected to electrocatalytic hydrogenation with a dual-catalyst Pt/C and polyoxometalate. Significant conversion of lignin monomers was achieved in all three lignin oils. The main products of the ECH of the poplar derived lignin oil were propylcyclohexane (hydrolysis product) and 4-propylcyclohexanol (hydrogenation product) achieving yields of 42.6% and 29.8%, respectively (Reaction conditions: 100 mA/cm^2^, 80 °C, 4 h reaction time, 0.25 M PW_12_, 5 mol% Pt/C). Peng et al. [72] studied the ECH of lignin oils derived from pinewood and birch-wood using PtRhAu catalyst [144]. A schematic of the process used by Peng et al. [72] is shown in Figure 16a. Their target product was 2-methoxy-4-propylcyclohexanol (2M4PC). The ECH of pinewood lignin oil (Figure 16b), which was composed mostly of 4-propylguaiacol, showed conversion to 2-methoxy-4-propylcyclohexanol with a 90% selectivity after 2 h of ECH. The results were similar for the ECH of birch wood lignin oil, which was composed of 4-propylsyringol (6.9 mM) and 4-propylguaiacol (1.8 mM), and was converted into 2-methoxy-4-propylcyclohexanol with 71% selectivity after 2 h of ECH (Reaction conditions: 50 mA cm^−2^, 0.2 M HClO4, room temperature).

## 5. Environmental and Economic Analysis

Electrochemical upgrading has the potential to significantly reduce the environmental impacts of the bio-oil thermochemical upgrading process particularly when coupled with electricity from renewable energy sources (i.e., wind or solar) [156,157]. Lam et al. [157] compared the energy, carbon, and mass yields of biochemical production of ethanol from cellulose with a combined pyrolysis-ECH system where ECH was used as a method for the stabilization of the pyrolysis oil before it was transported to central refinery for further hydrotreatment. Lam et al. found that the pyrolysis-ECH process retained 89% of the energy input into the process compared to only 44% for the biochemical cellulose to ethanol process. Although a full life cycle assessment (LCA) was not performed, the study revealed that the pyrolysis-ECH system produced less carbon dioxide than the cellulose to ethanol process. Tu et al. [156] performed an LCA on the ECO of glycerol to lactic acid coupled with ECH of lignocellulosic derived fast pyrolysis oil to renewable fuel. They found that when the processes are coupled and powered by renewable energy sources (i.e, wind or solar power) it could reduce the carbon dioxide emissions to lactic acid production by 57% compared to the thermochemical baseline.

Das et al. [39] performed a technoeconomic analysis (TEA) on the process of converting corn stover to biofuels using distributed pyrolysis-ECH systems (called depots) and a centralized hydrotreating refinery. The results revealed that the biofuel produced from the pyrolysis-ECH system combined with a centralized refinery had a minimum fuel selling price (MFSP) of $0.95 per liter of gasoline equivalent (GLE) compared to $0.98 per GLE for cellulosic ethanol derived from fermentation of the same feedstock. Figure 17 shows the sensitivity analysis performed by Das et al. on the pyrolysis-ECH-Hydroprocessing system, which shows that changes in electricity cost have the highest impact on MFSP. Orella et al. [158] performed a TEA analysis for the conversion of guaiacol to phenol using ECH. They found that high FEs were qualitatively more important in determining product cost than phenol selectivity. They predicted that at a current density of 200 mA cm^−2^ combined with phenol FE of 75% and HER FE of 20%, the production cost of phenol using ECH would be $0.42 kg^−1^ which is significantly lower than the current production cost of phenol at $1.30 kg^−1^. Wang et al. [73] performed a TEA on the production of methoxycyclohexanol from guaiacol using ECH and found that that the cost of methoxycyclohexanol at a plant gate levelized cost would be $63.5 kg^−1^ compared to the current market price of $431 kg^−1^.

## 6. Effect of Hydrogen Source on Hydrogenation Reactions

Understanding the environmental and economic viability of ECH reactions requires direct comparison with thermo-catalytic hydrogenation (TCH). While TCH has been explored at a range of temperatures from 20 °C to 400 °C, researchers interested in comparing ECH and TCH reaction rates utilize the low temperature range between 20–80 °C [51,65,76,110]. As previously discussed, ECH results using in situ hydrogen production whereas TCH requires an external source of hydrogen gas [51,65,76,110]. During TCH adsorbed hydrogen is generated through the dissociation of hydrogen gas on the surface of the catalyst. During ECH this hydrogen is produced through the reduction of an electrolyte. An ECH cell operating at a current density of 10 mA/cm^2^ produces the equivalent of 10 MPa of adsorbed hydrogen on the catalysts surface [148]. TCH and ECH reaction rates and activation energies of phenol are very similar. According to Sing et al. [51] the TOF for ECH and TCH of phenol was the same (~16 s^−1^) with a 4 kJ/mol difference in activation energies (29 kJ mol^−1^ for ECH and 33 kJ mol^−1^ for TCH). Activation energy results at similar reaction conditions for TCH (1 bar H_2_) and ECH (0.3 V vs. RHE, acetic acid buffer, 25 °C, Pt/C electrode) suggest that the elementary steps of the hydrogenation reaction are independent of H^+^ generation method [51,78]. ECH of benzaldehyde has been reported as having consistently higher reaction rates and lower activation energies than TCH of benzaldehyde [65,76]. Singh et al. [76] proposed that the increase activity is due to higher H surface coverage during ECH.

## 7. Conclusions and Outlook

The economic conversion of bio-oils to renewable fuels and chemicals remains a challenge for widespread use of biofuels. ECH serves as a promising alternative to the current traditional bio-oil stabilization techniques, because it offers several advantages including low operating temperatures (<80 °C), ambient pressures, and no external hydrogen source. This review summarized recent advances on the ECH of bio-oil model compounds and bio crudes. The literature has focused on developing efficient and selective catalysts for the hydrogenation of model compounds found in bio-oils. The tunable selectivity of ECH allows for the targeted production of value-added chemicals such as methoxy-cyclohexanes, cyclohexanone and furfuryl alcohol, among others. ECH has shown to increase both the stability and hydrogen content of pyrolysis oil. Preliminary economic and environmental analyses have also shown that ECH can provide economic and environmental benefits compared to traditional thermochemical bio-oil upgrading processes.

Of all the catalysts studied in the literature several stand out as state-of-the-art materials. The dual-catalyst system proposed by Liu et al. [47] consists of a suspended noble metal catalyst (Pt/C) with a water soluble polyoxometalate (SiW_12_), and provides one of the only reported routes toward deoxygenation of phenols. This system also provides the highest efficiency (>99%) at industrially relevant current densities up to 800 mA cm^−2^. Alternatively, the PtRhAu catalyst proposed by Peng et al. [72] has shown to suppress cleavage of guaiacol’s methoxy group allowing for the selective production of the platform chemical 2-methoxy-cyclohexanol at a 60% faraday efficiency. For base metal catalysts, Cu remains the most selective catalyst for the deoxygenation reactions with several studies reporting ~60% selectivity for hydrogenolysis of furfural to 2-methylfuran in highly acidic conditions [49,134]. Other base metal catalysts such as Cu_2_P have shown excellent selectivity and efficiency for the production of furfuryl alcohol but are generally still limited by their efficiency and ECH rates compared to traditional PGM catalysts [99].

However, there are several knowledge gaps in the ECH literature. Limited research has been performed on real bio-oils or even mixtures of bio-oil model compounds. Individual compounds react differently depending on their functionality and the reaction conditions selected, while bio-oils are complex mixtures of organic molecules with a variety of different functional groups. Therefore, studies performed with multiple organics are needed to assess catalyst performance and stability. Furthermore, compared to traditional upgrading techniques (i.e., hydrodeoxygenation) the quality of the fuels produced using ECH is much lower. Tuning the ECH reactions to promote deoxygenation would significantly improve the quality of renewable fuels. Another major challenge is the low solubility of bio-oils in the aqueous phase. Most studies focused on real bio-oils use either low bio-oil concentrations or the water-soluble fraction of bio-oils, which is only a portion of the whole. The addition of organic solvents to increase oil solubility reduces the ECH reaction rates [80]. Thus, this is an area that needs to be further explored.

ECH remains a promising technology for the future of biofuels production, but further research is needed prior to its commercialization. The mild operating conditions combined with the ability to directly utilize electricity produced from solar and/or wind power makes ECH of bio-oils an attractive technology for the production of sustainable and renewable carbon-based liquid fuels and chemicals. Widespread adoption of ECH could provide a pathway toward environmentally and economically favorable distributed production of biofuels.

## Figures and Tables

**Figure 1 materials-16-00394-f001:**
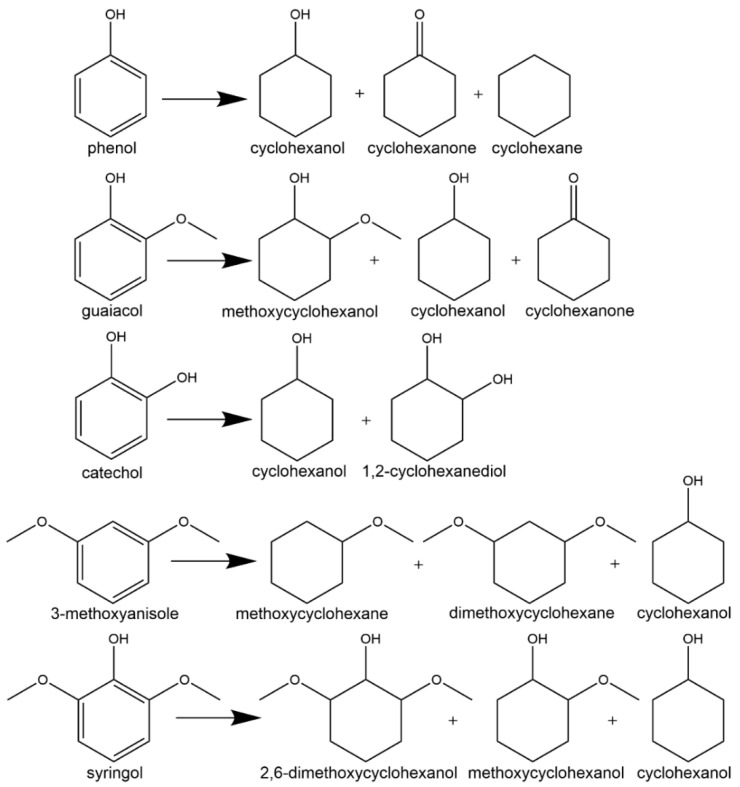
Reaction products observed after ECH of several bio-oil model compounds with different functional groups [47,60].

**Figure 2 materials-16-00394-f002:**
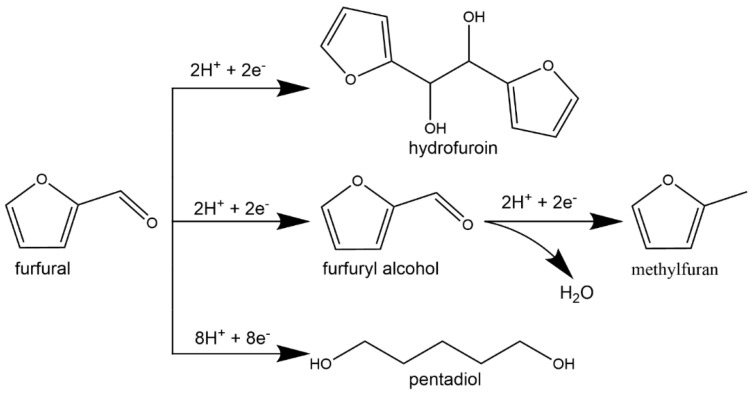
ECH reaction network of furfural hydrogenation. Adapted with permission from Reference [42]. Copyright 2022 American Chemical Society.

**Figure 3 materials-16-00394-f003:**
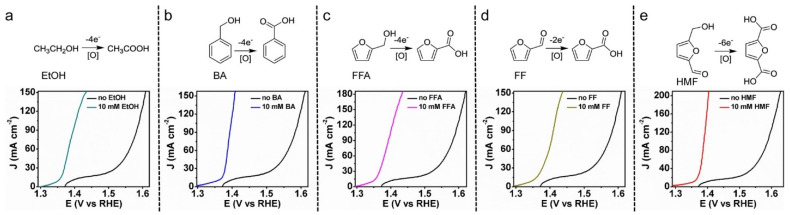
(**a**−**e**) Oxidation of selected important biomass-derived organics to value-added products and corresponding linear sweep voltammetry curves with or without 10 mM of the organic substrates (10 mM, pH = 14, scan rate of 2 mV s^−1^), using a nickel foam. Reprinted with permission from Reference [95]. Copyright 2016 American Chemical Society.

**Figure 4 materials-16-00394-f004:**
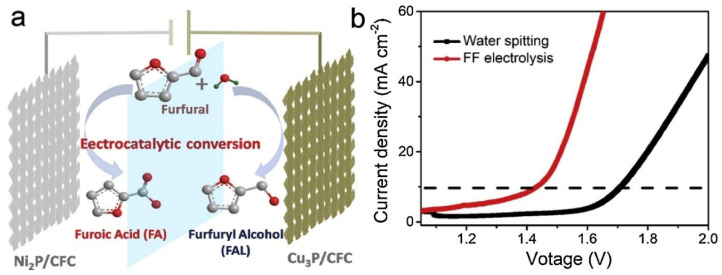
(**a**) Schematic diagram of Cu_3_P/CFC and Ni_2_P/CFC for two-electrode electrocatalytic conversion of furfural. (**b**) LSV plots without iR compensation of Cu_3_P/CFC (as cathode) and Ni_2_P/CFC (as anode) two-electrode system in 1.0 M KOH without and with 50 mM furfural. Reprinted with permission from Ref [99]. Copyright 2019 Elsevier Inc.

**Figure 5 materials-16-00394-f005:**
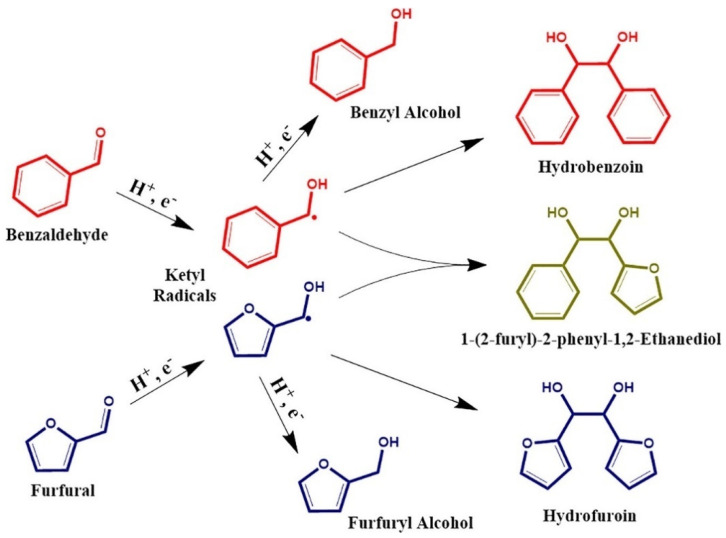
Schematic Representation of the Electrochemical Reduction Pathways for the Mixed Benzaldehyde—Furfural Reduction System. Reprinted with permission from Reference [62]. Copyright 2020 American Chemical Society.

**Figure 6 materials-16-00394-f006:**
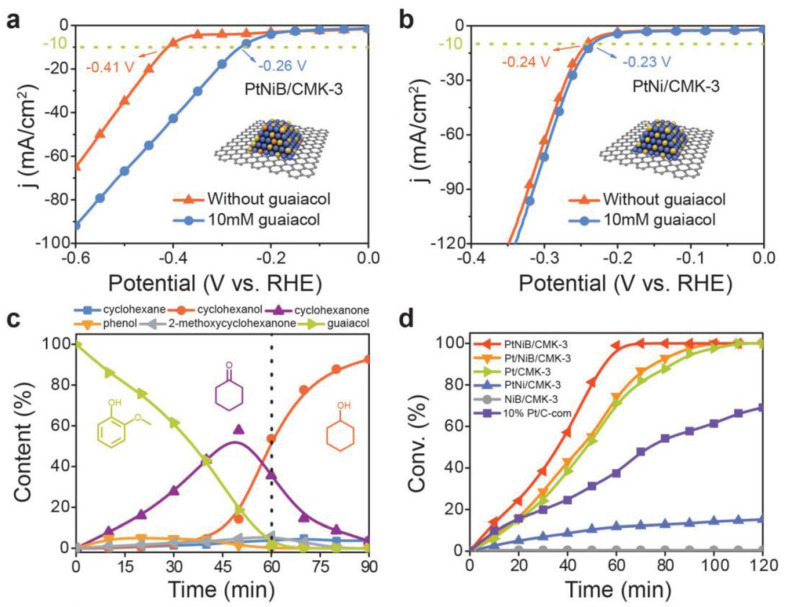
(**a**) HER and ECH polarization curves of PtNiB/CMK-3. (**b**) HER and ECH polarization curves of PtNi/CMK-3. (**c**) Content (%) of guaiacol and its hydrogenation products during ECH by PtNiB/CMK-3. (**d**) Conversion curves of guaiacol on ECH reaction for different electrocatalysts. Reproduced with permission from ref [43]. Copyright 2019 Wiley-VCH.

**Figure 7 materials-16-00394-f007:**
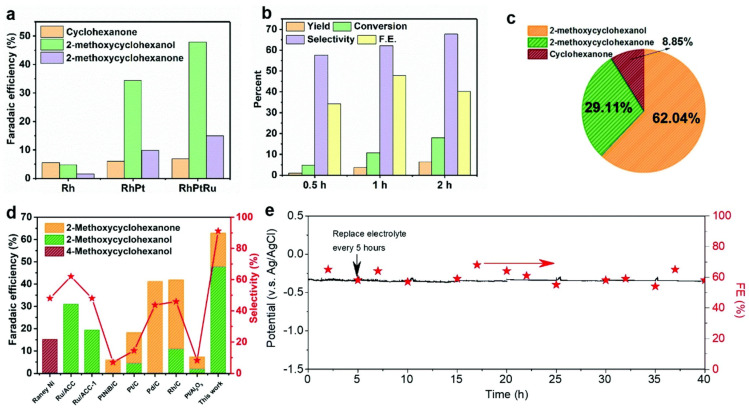
(**a**) Faradaic efficiency (FE) toward various hydrogenated products from guaiacol using Rh, RhPt, and RhPtRu catalysts at 50 mA cm^−2^ after 1 h electrocatalytic reaction. (**b**) Yield, conversion, selectivity, and FE towards 2-methoxycyclohexanol obtained by upgrading guaiacol using a RhPtRu catalyst at 50 mA cm^−2^ for up to 2 h. (**c**) Pie chart with the molar ratio of various hydrogenated products over the RhPtRu catalyst at 50 mA cm^−2^ after 1 h electrocatalytic reaction. (**d**) Comparison of Wang et al. [73] with representative research that targeted the preservation of oxygenated functional groups (OFGs, –OCH3) via ECH. Right-Y axis represents the selectivity of methoxylated products over various catalysts. (**e**) The ECH of guaiacol on RhPtRu catalysts at an applied current density of 50 mA cm^−2^ using a H-cell system with 50 mL of catholyte for 40 h, showing the potential and FE to the target methoxylated products. Note that the catholyte was replaced every 5 h. Reproduced with permission from ref [73]. Copyright 2022 The Royal Society of Chemistry.

**Figure 8 materials-16-00394-f008:**
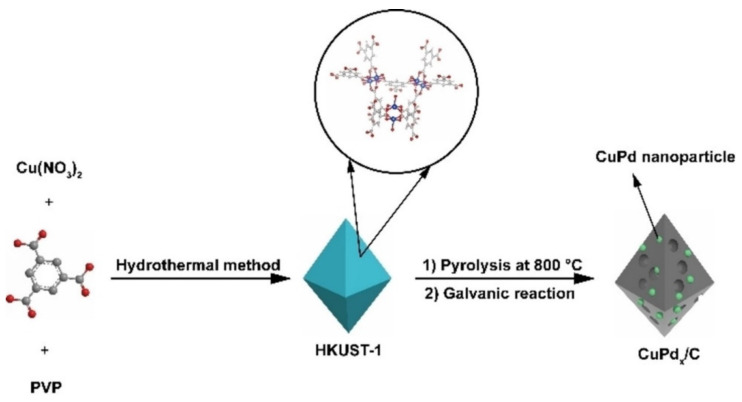
Schematic illustration of the method for the synthesis of CuPd_x_. Reproduced with permission from ref [126]. Copyright 2022 Wiley-VCH.

**Figure 9 materials-16-00394-f009:**
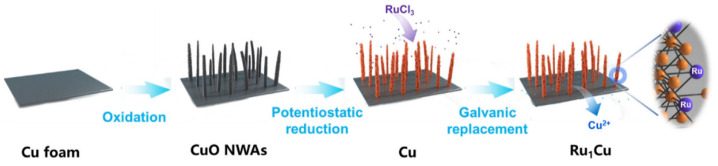
Schematic illustration of synthetic route for Ru_1_Cu single atom alloys. Reproduced with permission from ref [87]. Copyright 2022 Wiley-VCH.

**Figure 10 materials-16-00394-f010:**
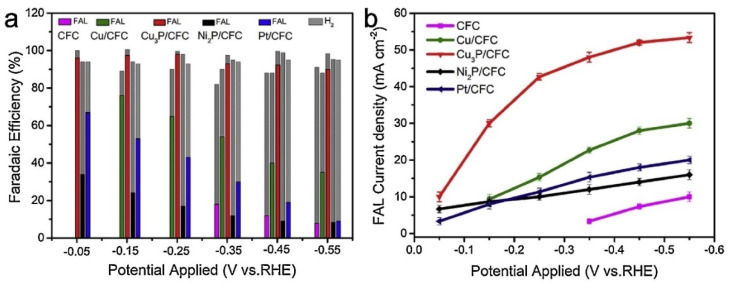
The ECH of furfural over different electrodes: (**a**) the FAL and H_2_ Faradaic efficiency, (**b**) the FAL partial current density at different applied potential measured in N_2_-saturated 1 M KOH solution with 50 mM furfural in the working electrode reaction tank. Reprinted with permission from Ref [99]. Copyright 2019 Elsevier Inc.

**Figure 11 materials-16-00394-f011:**
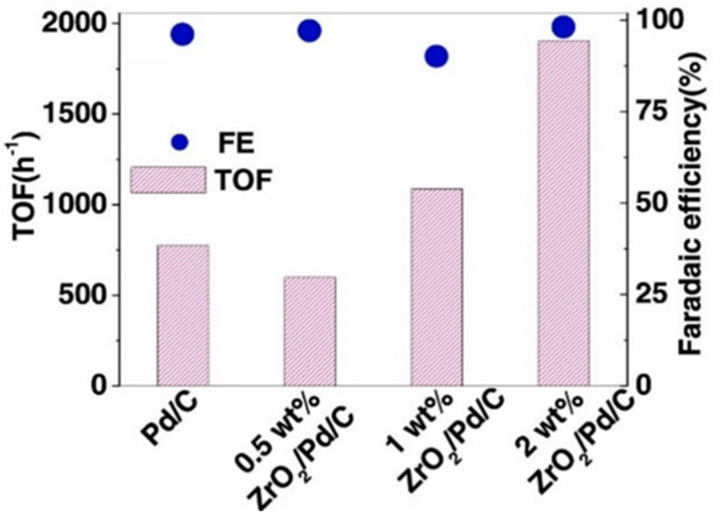
Electrocatalytic hydrogenation of benzaldehyde on Pd/C and ZrO_2_/Pd/C catalysts with different ZrO_2_ loadings. Reprinted with permission from Ref [121]. Copyright 2021 Elsevier Inc.

**Figure 12 materials-16-00394-f012:**
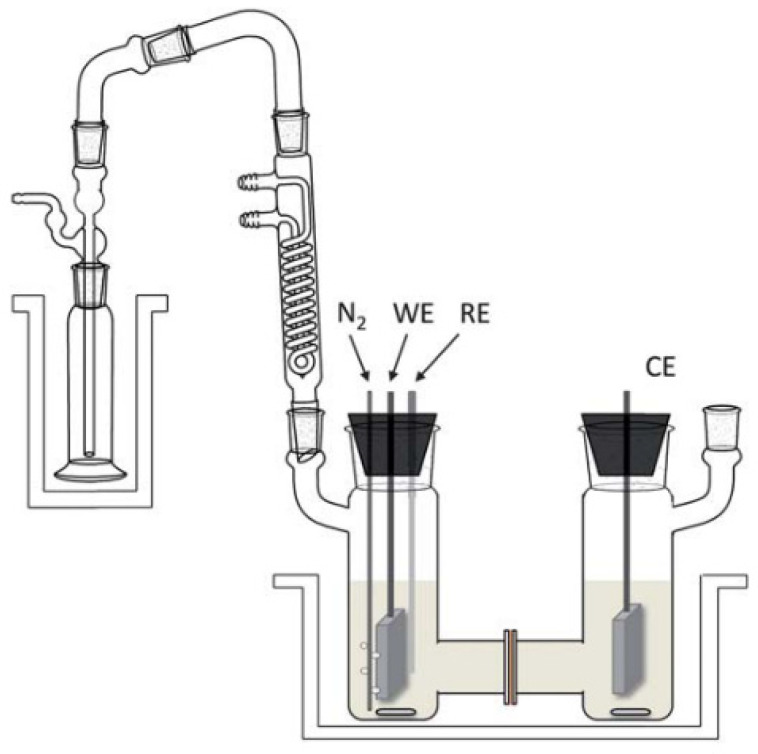
Electrochemical set-up with a divided H-cell, reflux condenser and cryo-trap; working electrode (WE), reference electrode (RE), counter electrode (CE). Reproduced with permission from ref [134]. Copyright 2013 The Royal Society of Chemistry.

**Figure 13 materials-16-00394-f013:**
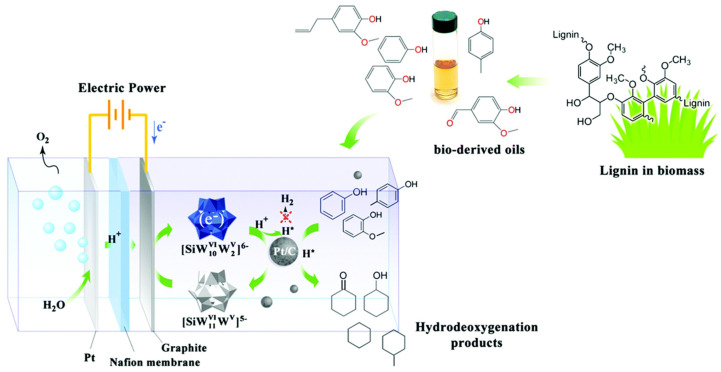
Dual-catalyst slurry system. Reproduced with permission from ref [47]. Copyright 2020 The Royal Society of Chemistry.

**Figure 14 materials-16-00394-f014:**
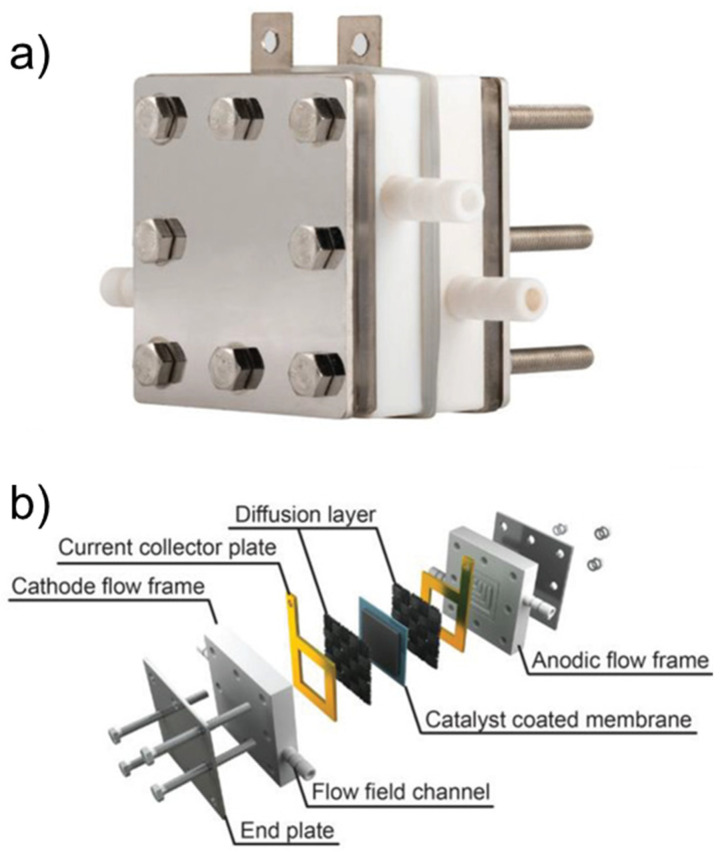
Flow ECH cell (**a**) assembled (**b**) expanded view of the structural components. Reproduced with permission from ref [43]. Copyright 2019 Wiley-VCH.

**Figure 15 materials-16-00394-f015:**
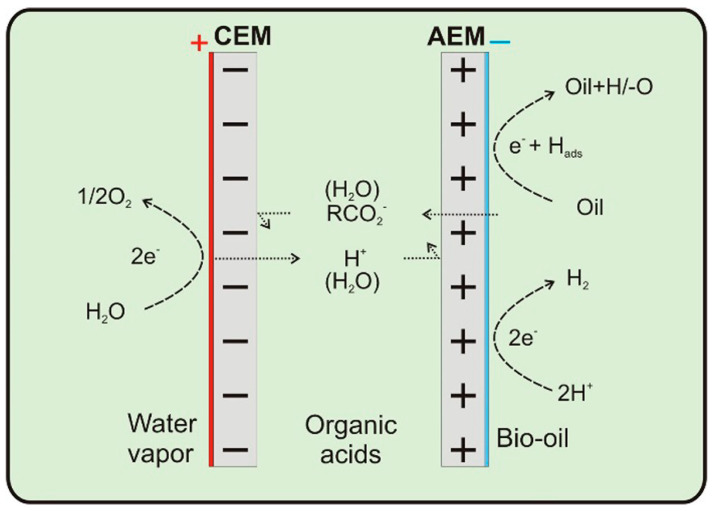
Spatial representation of the dual membrane electrolyzer cell. Reprinted with permission from Reference [153]. Copyright 2018 American Chemical Society.

**Figure 16 materials-16-00394-f016:**
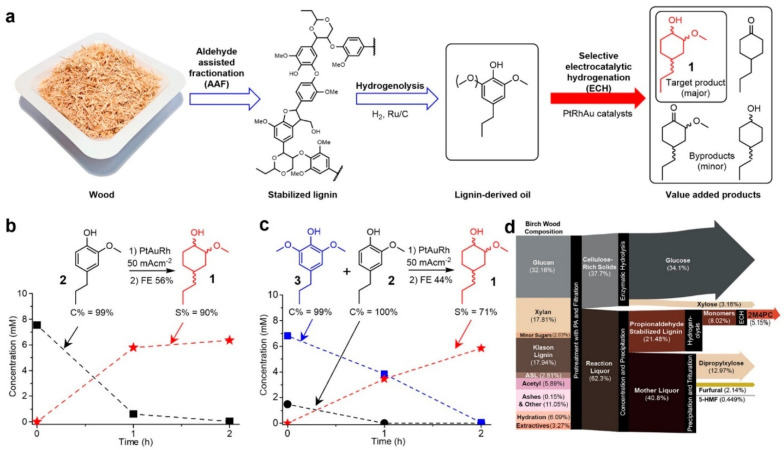
Integrated lignin valorization from wood to a methoxylated product 2-methoxy-4-propylcyclohexanol (2M4PC). (**a**) Integrated lignin valorization process for high-value 2M4PC from wood. 2M4PC (1, red) is the target structure in bold. (**b**) Concentration evolution of pinewood- derived lignin monomer 4-propylguaiacol (4PG) (2, black) and target product 2M4PC (1, red), showing an FE at 1 h reaction, conversion rate (C %), and product selectivity (S%) toward 2MC over 2 h ECH reaction. (**c**) Concentration evolution of birch-wood-derived lignin oil consisting of 4-propylsyringol (4PS) (3, blue) and 4PG (2, black) and target product 2M4PC (1, red), showing an FE at 1 h reaction, C%, and S% toward 2MC over 2 h ECH reaction. (**d**) Mass balances during the propionaldehyde-assisted fractionation of lignocellulosic biomass, as performed on birch wood. All the numbers are provided as weight percentages. The provided weight percentages of the sugars, stabilized sugars, furfural, 5-hydroxymethylfurfural (5-HMF), stabilized lignin, and lignin monomers have been corrected for the mass of the stabilizing group, hydration, dehydration, or hydrogenation to match their initial structure in the native biomass. Reprinted with permission from Reference [72]. Copyright 2018 American Chemical Society.

**Figure 17 materials-16-00394-f017:**
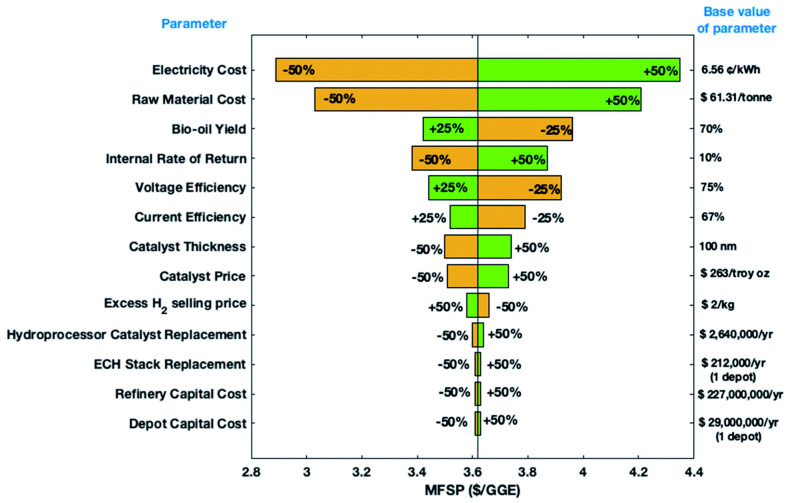
Tornado plot showing MFSP single parameter sensitivity analyses for the pyrolysis-ECH-Hydro-processing system (Depot size of 500 tons per day and refinery size of 2000 tons per day). Reproduced with permission from ref [39]. Copyright 2022 The Royal Society of Chemistry.

**Table 1 materials-16-00394-t001:** Summary of the literature on the ECH of bio-oil model compounds with platinum group metals.

Catalyst	Model Compound	Target Product (Selectivity %)	Electrolyte	Temp (°C)	TOF (h^−1^)	j (mA/cm^2^)	FE (%)	REF
PtNi/NHPC	Phenol	Cyclohexanone (99.9)	H_2_SO_4_	25	115.1	7.5	87.7	[43]
PtNiB/CMK-3	Guaiacol	Cyclohexanol (90.3)	HClO_4_	60	291.28	10	90.8	[108]
Ru/ACC	Guaiacol	Cyclohexanol (53)	HCl	80	NR	100 *	30	[52]
Pt/C	Phenol	Cyclohexanol (56)	H_2_SO_4_	25	77.7	7.5	NR	[43]
Pt/C	Phenol	Cyclohexanol (NR)	Acetate Buffer	25	270	100 *	19	[58]
Rh/C	Phenol	Cyclohexanol (NR)	Acetate Buffer	25	296	100 *	68	
Pd/C	Phenol	Cyclohexanol (NR)	Acetate Buffer	25	<0.002	40 *	<1	[50]
Pt-Graphite	Phenol	Cyclohexane (63%)	H_2_SO_4_, HClO_4_, HCl	80	NR	30	23 ^	[68]
Pt/C	Benzealdehyde	Benzyl alchol (NR)	Acetate Buffer	RT	2189	200 *	39	[65]
Rh/C	Benzealdehyde	Benzyl alchol (NR)	Acetate Buffer	RT	2267	185 *	64	
Pd/C	Benzealdehyde	Benzyl alchol (NR)	Acetate Buffer	RT	3899	300 *	99.6	
Pt/C	Furfural	Furfuryl alchol (NR)	Acetate Buffer	RT	250	0.7 V	NR	[58]
PdAg/C	Furfural	Furfuryl alchol (95)	Na_3_PO_4_	60	NR	0.5 V	>95%	[48]

* reported as mA, ^ reported as FE to cyclohexane, (NR) not reported, (RT) room temperature.

## Data Availability

Not applicable.

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
