# Peer review of "Recent Progress in Electrochemical Upgrading of Bio-Oil Model Compounds and Bio-Oils to Renewable Fuels and Platform Chemicals"

_materials, 2023, doi:10.3390/ma16010394_

Round 1

Reviewer 1 Report

Current manuscript entitled “Recent Progress in Electrochemical Upgrading of Bio-oil Model Compounds and Bio-oils to Renewable Fuels and Platform Chemicals” by “Page et al” reviewed on the on bio-oil upgrading using ECH. Factors impacting ECH reactions are systematically presented and analyzed, including electrode design, reaction temperature, applied overpotential, and electrolytes. This review finds ECH a promising avenue to produce renewable, carbon-based, drop-in biofuels as the world transitions to electricity produced from wind and solar power and summarizes the challenging tasks that we need to undertake to realize the technology on an industrial scale. The manuscript seems good and can be accepted after addressing the following comments.

1.      In the introduction authors should discuss on the exiting reviews and literature gap of the current review.

2.      Provide the challenges that are currently facing with the Electrochemical Upgrading Reactions

3.      It will be good if the authors provide brief conclusions of the work, it will be easy for the readers.

4.      In the manuscript, please place the references before “.”. Revise this throughout the manuscript.

5.      In the introduction provide information on the Electrochemical hydrogenation/Hydrogenolysis.

Reviewer 2 Report

Title: Recent Progress in Electrochemical Upgrading of Bio-oil Model Compounds and Bio-oils to Renewable Fuels and Platform Chemicals

Summary: Sustainable production of renewable carbon-based fuels and chemicals remains a necessary but immense challenge in the fight against climate change. Present review finds ECH a promising avenue to produce renewable, carbon-based, drop-in biofuels as the world transitions to electricity produced from wind and solar power and summarizes the challenging tasks that we need to undertake to realize the technology on an industrial scale. The review is very well constructed and supported by enough reference evidence. The following are my minor suggestions for further improvement of the review.

Comment:

1.       Abstract can be more impressive, include the review topic focused or targeted points.

2.       The novelty of the review can be included in abstract with single line.

3.       The abbreviations need to be predefined.

4.       The first line of the introduction can be cited ‘Over the past several decades the utilization of renewable energy has attracted significant interest worldwide’ with previously published research such as Exploitation of cost-effective renewable heterogeneous base catalyst from banana (Musa paradisiaca) peel for effective methyl ester production from soybean oil.

5.       Objectives need to be more precise.

6.       Effect of reaction conditions on the lifetime of SAPO-34 catalysts in methanol to olefins process–A review could be helpful in the introduction section.

7.       Table 1 needs to format in the text format not in the image format.

8.       Authors provided several figures with proper copyright permissions. But only 1 table is included for Electrode and Catalyst Support Materials/ Flow type cells. The comparative table can be considered with a wide range of literature surveys.

9.       Units needs to be standardized and superscripts need to be checked.
